# Methods and Challenges of Using the Greater Wax Moth (*Galleria mellonella*) as a Model Organism in Antimicrobial Compound Discovery

**DOI:** 10.3390/microorganisms7030085

**Published:** 2019-03-19

**Authors:** Athina Andrea, Karen Angeliki Krogfelt, Håvard Jenssen

**Affiliations:** 1Department of Science and Environment, Roskilde University, 4000 Roskilde, Denmark; atan@ruc.dk (A.A.); kak@ssi.dk (K.A.K.); 2Virus and Microbiological Special Diagnostics, Statens Serum Institut, 2300 Copenhagen, Denmark

**Keywords:** invertebrate model, *Galleria mellonella*, infection model, antimicrobial compound

## Abstract

Among non-mammalian infection model organisms, the larvae of the greater wax moth *Galleria mellonella* have seen increasing popularity in recent years. Unlike other invertebrate models, these larvae can be incubated at 37 °C and can be dosed relatively precisely. Despite the increasing number of publications describing the use of this model organism, there is a high variability with regard to how the model is produced in different laboratories, with respect to larva size, age, origin, storage, and rest periods, as well as dosing for infection and treatment. Here, we provide suggestions regarding how some of these factors can be approached, to facilitate the comparability of studies between different laboratories. We introduce a linear regression curve correlating the total larva weight to the liquid volume in order to estimate the in vivo concentration of pathogens and the administered drug concentration. Finally, we discuss several other aspects, including in vivo antibiotic stability in larvae, the infection doses for different pathogens and suggest guidelines for larvae selection.

## 1. Introduction

In order to study infectious diseases, as well as the in vivo potential of novel antimicrobials, developing reliable animal models that mimic clinical manifestations in humans is essential. Mammalian models are popular and often unanimously irreplaceable due to their similar physiology to humans. Nevertheless, high cost and ethical issues complicate their use, especially when it comes to large-scale settings. Invertebrate models offer a real possibility as early-stage in vivo screening models, in an attempt to lower the number of candidate compounds proceeding to further evaluation in mammalian models [1]. Invertebrates are excluded from the Animals Act 1986; therefore, ethical constraints are relatively limited. Even though insects lack an adaptive immune system, they have an innate immune system similar to that of mammals [2,3,4].

Among insect models, the caterpillar larva of the greater wax moth *Galleria mellonella* (order Lepidoptera, family *Pyralidae*) offers several attractive advantages, including the ability to incubate at 37 °C, short life span, a size that is convenient for handling, and relatively precise dosing. The wax moth’s popularity has remarkably increased in the last decade, with a plethora of studies providing evidence that the *Galleria mellonella* (*G. mellonella*) is a reliable model to test pathogenicity of bacteria, fungi, and viruses, as well as testing the effectiveness and toxicity of antimicrobial compounds [5,6,7]. When comparing different studies, it is apparent that there is a profound discrepancy from laboratory to laboratory, with respect to how the larva infection model is performed (Table 1). Variability is due to several factors, including, among others, larvae sizes, origin, storage, dosing of infection, and treatment. The aim of the present work is to propose practices in an attempt to unify larvae infection protocols so that it will become easier to compare data across different studies in the future.

## 2. Materials and Methods

### 2.1. Galleria mellonella Larvae Liquid Volume Determination

Last instar larvae were purchased from a local vendor (MiniZoo, Copenhagen, Denmark). Individual healthy larvae of different sizes (*n* = 66) were weighed and placed in 1.5 mL eppendorf tubes with a hole drilled on their lid. The weight of each individual tube containing a single larva was also determined. Larvae were allowed to freeze at −80 °C overnight and were subsequently freeze–dried for three days (Alpha 1-2 LDplus, Martin Christ Gefriertrocknungsanlagen GmbH) using a vacuum pump (RV12, Edwards). The weight of the tubes containing the larvae was measured, and the weight of the liquid was determined by subtracting the weight of the tube before and after drying. The haemolymph weight to volume ratio was found to be approximately one; therefore, we accepted that the liquid weight of a single larva approximated its volume. The larva weight in milligrams was plotted against the larva liquid volume in microliters. The regression line and R square values were calculated using Excel.

### 2.2. Bacteria and Growth Conditions

For *G. mellonella* infection experiments, the following bacterial strains were used: *Pseudomonas aeruginosa* ATCC27853 (*P. aeruginosa*), *S. aureus* ATCC29213, *Escherichia coli* ATCC25922 (*E. coli*), *Staphylococcus epidermidis* (*S. epidermidis*) clinical strain (HJ056) [21], and a methicillin–resistant *S. aureus* (WKZ-2) [22]. All strains were cultivated in Mueller–Hinton broth (MHB) (BD Difco, DF0757-07-8) or agar (MHB with 1.5% agar). Cultures were incubated at 37 °C with shaking at 180 rpm.

### 2.3. Galleria mellonella Infection

Last instar larvae were purchased from a local vendor (MiniZoo, Copenhagen, Denmark). Upon arrival, healthy larvae of 200 to 300 mg were selected (larva selection guidelines are reported in Appendix A) for the experiments and were stored at 13 °C for up to 2 weeks, without food. Larvae were allowed to acclimatize at 37 °C one day before the experiment. Overnight cultures of bacteria were prepared in 5 mL MHB. One milliliter of each culture was centrifuged for 7 min at room temperature at 6700× *g*. Supernatants were discarded, and the pellet was re-suspended in 1 mL of phosphate buffered saline (PBS). The bacterial suspension was diluted in PBS to an OD_600nm_ of 1, which corresponded to approximately 1 × 10^9^ colony forming units per mL (CFU/mL). Inocula were always confirmed by viable counts. Groups of 10 larvae were topically disinfected at the last left proleg with 70% ethanol using a cotton swab and were injected with 10 μL bacterial suspension appropriately diluted in PBS, using an Insumed syringe (0.3 mL, 31 G, Pic solutions). For each experiment, larvae injected with 10 µL PBS served as a control group to ensure that the injection procedure was not causing death. Larvae were incubated at 37 °C in the dark, without food, and were considered dead when they were unresponsive to touch and dark brown to black in color. Survival was monitored every 24 h for up to four days.

### 2.4. Bacterial Load in the Haemolymph

Infected larvae were collected, and their surfaces were disinfected with 96% ethanol. The larvae were placed on ice until no movement was observed. The haemolymph was collected individually from each larva in eppendorf tubes by holding the larva with tweezers and puncturing the cuticle with a sterile 18 G needle behind the head. The haemolymph was allowed to drain in the eppendorf tube. Immediately after, serial 10-fold dilutions of the haemolymph were prepared in PBS. Ten microliters of each dilution were plated on MH agar, in triplicates, and were allowed to dry. Colonies were counted manually the next day after an overnight incubation at 37 °C, and the CFU/mL haemolymph was calculated. No colonies were obtained from control, non–infected larvae.

### 2.5. In Vivo Stability of the Antibiotic

To investigate the in vivo stability of the antibiotic, healthy larvae of 350–400 mg were injected with 490 µg/mL ciprofloxacin (Sigma, 17850). Larvae injected with PBS served as a control. Larvae were incubated at 37 °C, and at different time points (10 min, 1, 3, and 24 h), the haemolymph was extracted (individually for each larva), as described in Section 2.4. Ten microliters of the freshly extracted haemolymph of ciprofloxacin- or PBS-injected larvae were placed on MH agar, for each time point, and were allowed to dry. One hundred microliters of an overnight *E. coli* culture were spread over the agar to create a layer of bacteria. The plates were incubated overnight at 37 °C, and inhibition zones were observed the next day. The experiment was repeated twice, and representative plates are presented.

## 3. Results

### 3.1. A Linear Regression Curve for G. mellonella Larvae Liquid Volume Determination

Estimating the liquid volume of a larva can be important for dosing calculations. Often treatment doses are expressed as milligrams of a compound per larva kilogram (mg/kg), following the human dosing recommendations (Table 1). However, for compounds for which the minimum inhibitory concentration (MIC) value is available, it can be relevant to express the treatment as Y× MIC in vivo. Moreover, estimating the in vivo concentration of pathogens (CFU/mL of larva liquid volume) can provide valuable information in infection experiments and allow for comparison between studies where larvae of different sizes are employed. In order to correlate the total weight of a larva to its liquid volume, we used 66 larvae of various sizes (weight range 100–740 mg). Larvae were dried, and the liquid volume was plotted against the larva total weight (Figure 1). The liquid volume was found to have a good correlation with the total larva weight, and the regression line gave an *R* square value of 0.995.

To validate the standard curve, we run two simple experiments. In the first experiment, larvae (minimum of 3 larvae per weight group) of four different weight ranges (100–140, 200–240, 300–340, and 500–540 mg) were infected with previously known lethal and non-lethal concentrations of *E. coli*-(100% lethal or non-lethal, respectively, after 24 h). The previously known lethal/non-lethal concentrations were determined for larvae with a weight of 400–500 mg and were found to be 8 × 10^6^ CFU/larva and 8 × 10^4^ CFU/larva, respectively. The average theoretical larva liquid volume for this weight range was calculated from the curve to be 284 µL; therefore, the theoretical in vivo concentration was calculated for the lethal and non-lethal doses to be 3 × 10^7^ and 3 × 10^5^ CFU/mL larva liquid, respectively. The appropriate bacterial suspensions were prepared in order to achieve this theoretical in vivo concentration immediately after infection. Inocula were confirmed by viable counts. After 24 h, the larvae receiving a lethal dose had 100% mortality, while all the larvae groups receiving the non-lethal dose had 100% survival (Figure 2). In the very low weight group (100–140 mg) with the larvae receiving a non-lethal infection, there was one dead larva; however, this can be attributed to the injection procedure itself as these larvae were very small. This finding supports that the curve is fit for use.

For the second experiment, larvae (*n* = 4) were infected with a known concentration of *E. coli*-(confirmed by viable counts) and were sacrificed immediately after infection. The in vivo bacterial load was determined in the haemolymph by viable counts and was compared to the theoretical CFU/mL larva liquid as calculated from the curve. The theoretical versus the measured in vivo CFU/mL larva liquid was very similar for all four tested larva weight ranges (Figure 3), which indicates that the larva liquid volume, as calculated from the curve, approximated reality. Control larvae injected with PBS had no bacteria in their haemolymph, as proved by plating.

### 3.2. Infection and Treatment Dose Must Be Tuned

Infection and treatment doses can be tuned using the larva liquid volume equation presented in Figure 1. Different bacterial strains demonstrated different pathogenicity in *G. mellonella* larvae. For example, for the *P. aeruginosa* strain used in the present study, the lethal dose after 24 h was only 50 CFU/larva, while for the *E. coli* strain, the lethal dose was in the magnitude of a million CFU/larva (Figure 4). Therefore, the infection dose must be tuned for each pathogen. For antimicrobial compound efficacy studies, we believe that larvae should be infected with the minimum lethal dose after 24 h. A higher infection dose might lower the chances of a candidate antimicrobial compound to work, and 24 h gives a relatively fast screening. An example of how to prepare the injected solution to achieve a specific in vivo concentration is presented in Figure 5.

Another parameter for consideration when it comes to treatment doses is the lifetime of each test compound in vivo. To find evidence about how long the antibiotic is active in vivo, *G. mellonella* larvae 350–390 mg were injected with 490 µg/mL ciprofloxacin solution. At different time points (10 min, 1, 3, and 24 h), the haemolymph was collected from individual larvae. Haemolymph drops were plated on agar plates, and a loan of fresh *E. coli* culture was deposited on top. The next day, inhibition zones were evaluated, and interestingly there were no inhibition zones at 24 h post-administration (Figure 6). This finding might suggest that larvae-specific parameters inactivate ciprofloxacin in vivo after 24 h. Therefore, re-administration of tested compounds has to be considered for longer experiments.

## 4. Conclusions

*G. mellonella* larvae are a reliable, cheap, and immunologically complex model for the preliminary in vivo investigation of the therapeutic potential of novel antimicrobial compounds, which precede mammalian models. However, the outcomes of this model are highly influenced by the methodology used and the larva’s age; therefore, an attempt should be made to standardize the research practices for this model. In this work, we presented a linear regression curve that correlates the larva total weight with the liquid volume. This curve can be used as a tool by *G. mellonella* model researchers in order to estimate the in vivo concentration of pathogens and administered compounds and will facilitate the comparison of results across different studies.

## Figures and Tables

**Figure 1 microorganisms-07-00085-f001:**
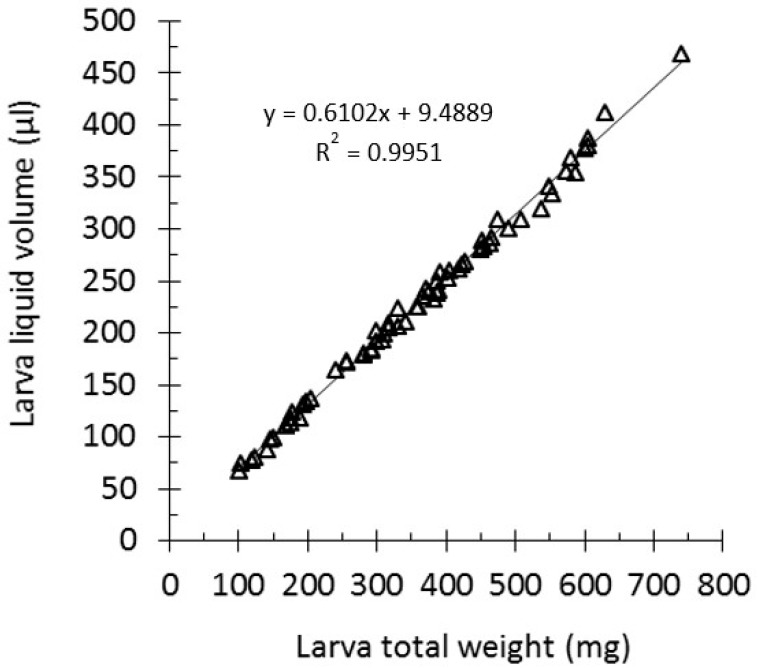
*G. mellonella* larva total weight versus the liquid volume curve. Healthy larvae of different sizes (*n* = 66) were weighed and freeze-dried. The wet-minus-dry weight was the larva liquid weight, which was found to be almost identical to the liquid volume. Each triangle represents data from a single larva. The regression line was drawn, and the coefficient of determination *R* square was calculated in Excel.

**Figure 2 microorganisms-07-00085-f002:**
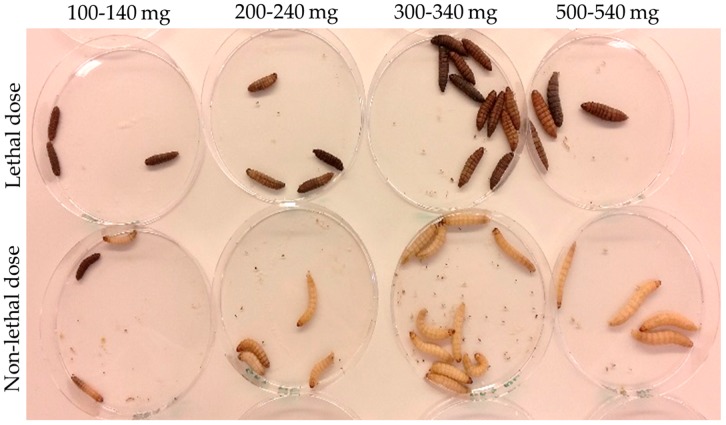
Lethal and non-lethal infection doses for different sizes of larvae. Larvae of different weights were infected with a known lethal and a known non-lethal concentration of *E. coli*. These concentrations were obtained in vivo based on the liquid volume calculation equation presented in Figure 1. The theoretical concentrations of the lethal and non-lethal cultures were confirmed to have the correct number of bacteria based on standard CFU counts. Larvae of different weights receiving a lethal dose had 0% survival after 24 h, while the larvae receiving a non-lethal dose survived.

**Figure 3 microorganisms-07-00085-f003:**
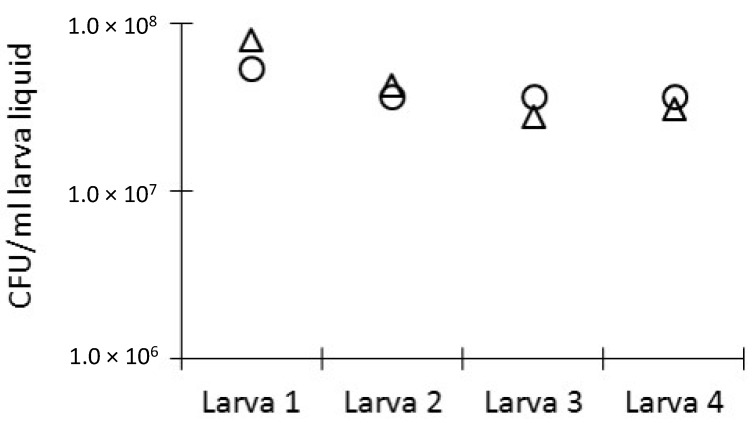
In vivo bacterial load in *G. mellonella* larvae. Larvae were infected with a known concentration of *E. coli* equal to 1.3 × 10^7^ CFU/larva, as confirmed by plate counting. The theoretical in vivo bacterial concentration was calculated based on larva liquid volume calculated from the curve. Immediately after infection, larvae were sacrificed and bacterial load was determined in their haemolymph by viable counts. The theoretically calculated (○) in vivo concentration was very similar to the measured (∆) concentration.

**Figure 4 microorganisms-07-00085-f004:**
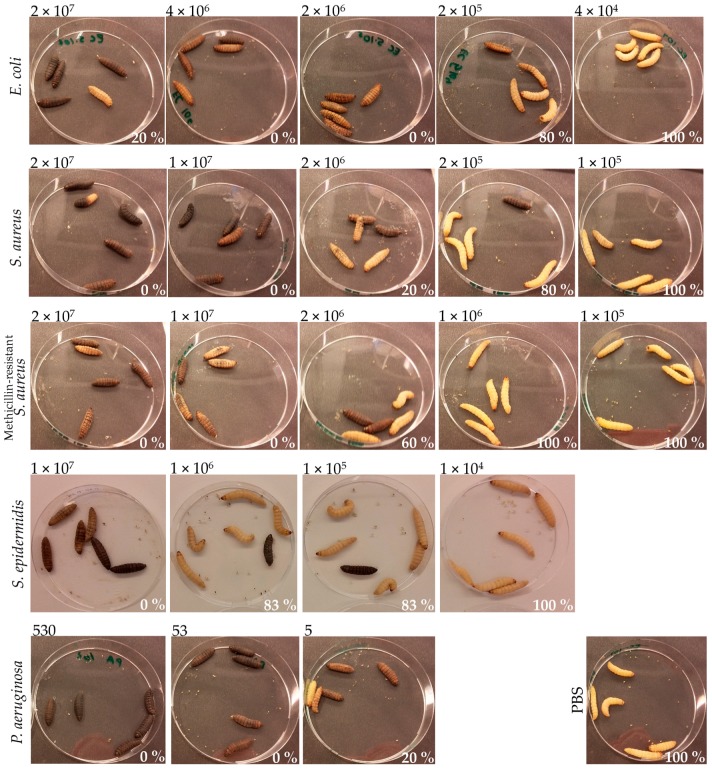
Pathogenicity of different bacterial species in *G. mellonella* larvae at 24 h post-infection. Larvae (*n* = 5–6) were infected with decreasing concentrations of each of the following pathogens: *E. coli*, *S. aureus*, methicillin–resistant *S. aureus*, *S. epidermidis*, and *P. aeruginosa*. Larvae injected with PBS served as a control and showed 100% survival (bottom right corner). The bacterial inoculum (CFU/larva), for each infection group, is presented on top of each picture. The survival (%) of the different infection groups 24 h post-infection is presented in the bottom right part of each picture.

**Figure 5 microorganisms-07-00085-f005:**
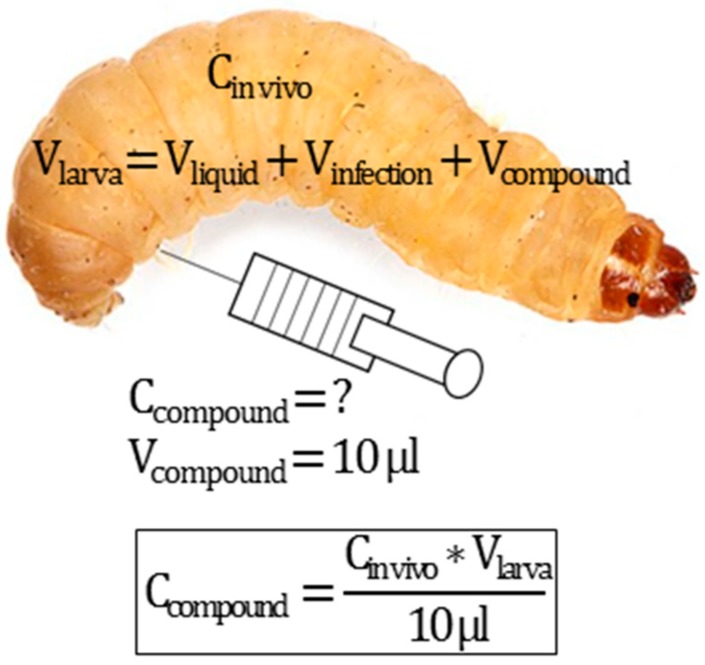
Example of the calculation of the concentration of the injected compound solution. In cases where a specific in vivo concentration is required (C_in vivo_), the concentration of the injected compound solution (C_compound_) can be calculated, for a specific volume of injection (V_compound_, typically 10 µL) and for any larva volume (V_larva_). The V_larva_ is the sum of the larva liquid volume (V_liquid_), as calculated from the curve presented in Figure 1 and the injection volumes (typically two injections, one for the infection V_infection_ and one for the treatment, 10 µL each).

**Figure 6 microorganisms-07-00085-f006:**
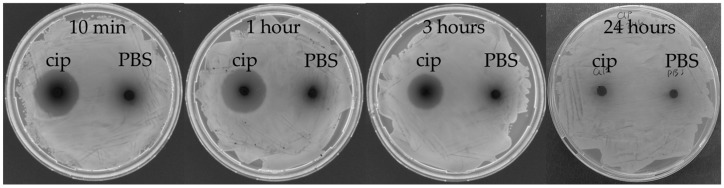
Antibiotic stability in vivo in *G. mellonella* larvae haemolymph. Ciprofloxacin (cip) at 490 µg/mL was injected in healthy larvae and after 10 min, 1, 3, and 24 h; the haemolymph was collected from one larva. Drops of 10 µL haemolymph were plated on agar and left to dry, after which a loan of *E. coli* was streaked on top. Inhibition zones were observed the next day. Larvae injected with PBS served as a control. Inhibition zones formed around the haemolymph spot from the antibiotic-injected larvae at 10 min, 1 and 3 h post-injection, but were not present at 24 h. The control haemolymph from the PBS-injected larvae did not form inhibition zones.

**Table 1 microorganisms-07-00085-t001:** Comparison of experimental aspects of the *G. mellonella* model infection protocol, as performed in different laboratories, for two bacterial pathogens (*Acinetobacter baumannii*, *Staphylococcus aureus*).

Experimental Aspect	Larva Size	Larva Origin	Larva Storage	Injection Site	Infection Dose CFU/larva	Antimicrobial Dose, Expressed as	Ref.
***A. baumannii***	250–350 mg	P-China	4 °C, 7 days	LP	1 × 10^5^	mg/larva kg	[8]
	Not given	P-UK	15 °C	Left proleg	1 × 10^4^	Not relevant	[9]
	250–350 mg	P-USA	7 days	LP	5 × 10^5^	mg/larva kg	[10]
	Not given	P-UK	RT, 14 days	FP	1 × 10^5^, 1 × 10^4^	Not relevant	[11]
	250 ± 25 mg	P-UK	15 °C	Not given	1 × 10^5^	mg/larva kg	[12]
***S. aureus***	150–200 mg	Reared	30 °C	Not given	1 × 10^6^	Not relevant	[13]
	Not given	P-UK	4 °C	Between segments	~1.3 × 10^6^	mg/larva kg	[14]
	15–25 mm long	P-UK	7 days	LP	0.8–2.6 × 10^6^	Not relevant	[15]
	~250 mg	P-China	Not given	LP	~1 × 10^6^	mg/larva kg	[16]
	300–700 mg	P-Netherlands	Not given	LP	1 × 10^6^	Not relevant	[17]
	~200 mg	Not given	Not given	LP	1 × 10^3^	µg/100 mg larva	[18]
	200–300 mg	P-USA	4 °C, 14 days	LP	5 × 10^6^	mg/larva kg	[19]
	Not given	Reared	28 °C	Not given	Not given	Not relevant	[20]

P = purchased, RT = room temperature, LP = last proleg, FP = front proleg, CFU = colony forming units, Ref. = Reference, *A. baumannii = Acinetobacter baumannii*, *S. aureus = Staphylococcus aureus.*

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
