# Peer review of "Methods and Challenges of Using the Greater Wax Moth (Galleria mellonella) as a Model Organism in Antimicrobial Compound Discovery"

_microorganisms, 2019, doi:10.3390/microorganisms7030085_

Round 1
Reviewer 1 Report
The author described how the variability in testing can be narrowed down from research paper to other. This is very significant as this makes a the research data more reliable and trust worthy.Few revisions
1) conclusion must be added to the paper.
2) Line 124, 127. figure # should be in order.
3) Figure 4, MRSA wording should be in italics
4) ul should be used in similar font across paper. In figure 5, once it is written in italics and once in normal font. Font of the entire paper should match. Looks like figure have different font wording than the text.
5) If you are using MIC,PBS or any other words in abbreviation forms; please specify the full form when it is first used in the paper; later on abbreviation should be good.
Good luck
Thanks
Author Response
Response to Reviewer 1 Comments
Point 1: conclusion must be added to the paper.
Response 1: Thank you very much for this comment. It was a big omission that we did not write a conclusion section. Conclusion has been added (lines 189-197): “G. mellonella larvae are a reliable, cheap and immunologically complex model for the preliminary in vivo investigation of therapeutic potential of novel antimicrobial compounds, before proceeding to mammalian models. However, the outcomes of this model are highly influenced by the methodology and larva age, therefore an attempt should be made to standardize the research practices for this model. In this work, we presented a linear regression curve that correlates larva total weight with liquid volume. This curve can be used as a tool by G. mellonella model researchers in order to estimate the in vivo concentration of pathogens and administered compounds and will facilitate the comparison of results from different studies.”
Point 2: Line 124, 127. figure # should be in order.
Response 2: Figure 1 is referenced in Figure 2 legend. Maybe the confusion arouses because Figure 1 is written in bold in the legend of Figure 2. This has been changed from bold to normal text.
Point 3: Figure 4, MRSA wording should be in italics
Response 3: “MRSA” has been replaced by “methicillin-resistant S. aureus” throughout the text.
Point 4: ul should be used in similar font across paper. In figure 5, once it is written in italics and once in normal font. Font of the entire paper should match. Looks like figure have different font wording than the text.
Response 4: Thank you for this comment, we had not noticed that. Figure 5 has been replaced by updated figure, “µl” in italics has been changed to normal text. Figure 2, Figure 4 and Figure 6 have been replaced by updated versions, where font has been changed to match the text (font “Palatino Linotype”). However, in Figure 5 “Cambria Math” could not be changed because the equation function is used. We hope this is ok.
Point 5: If you are using MIC,PBS or any other words in abbreviation forms; please specify the full form when it is first used in the paper; later on abbreviation should be good.
Response 5: Thank you for highlighting this, we had not noticed the full forms were missing. We have written the full form for both MIC (line 108) and PBS (line 76).
Reviewer 2 Report
Review of the article entitled: “Methods and challenges of using the Greater Wax Moth (Galleria mellonella) as a model organism in antimicrobial compound discovery”
Manuscript ID: microorganisms-460704
I have read the manuscript with great interest. In my opinion the results obtained by the author can be very helpful for other researchers who are going to use Galleria mellonella for investigation (in vivo) of virulence of selected pathogenic microorganism or therapeutic potential of novel antimicrobials. I think that the methods proposed by the authors of the manuscript could be considered as a base for preparing a final protocol which would be obligatory for all researchers who use Galleria mellonella for their tests. Of course I realize that many issues still have to be investigated and clarified. However, at the moment the authors perform their research using completely different conditions and it is difficult to compare the obtained results – Table 1. I my opinion the manuscript represent a good scientific quality and it deserves presenting as a publication in “Microorganisms”. However, before final acceptance some modifications would be required - the detailed comments are presented below
Abstract – well written no comments
Introduction – in general the introduction is informative and well written. I would only suggest writing it clearly – emphasize (in the first paragraph) that the animals such as Galleria mellonella can be used instead of mammalian models only for preliminary tests.
Materials and Methods – the authors should give the source of both clinical strains of bacteria (S. epidermidis HJ014 and MRSA). Moreover, in this part of the manuscript authors omitted the research aiming in investigation of stability of antibiotic – they only presented it in the paragraph “Results”. I also would be grateful for a short information (one sentence) how the authors performed counting of bacteria.
Results – well written and very interesting.
Conclusions – the authors should present short but separated conclusion.
My final decision is – minor revision.
Author Response
Response to Reviewer 2 Comments
Point 1: Introduction – in general the introduction is informative and well written. I would only suggest writing it clearly – emphasize (in the first paragraph) that the animals such as Galleria mellonella can be used instead of mammalian models only for preliminary tests.
Response 1: Thank you for pointing out that this has not been clear. We have now stated it more clearly in the new manuscript version (lines 30-32): “Invertebrate models offer a real possibility as early-stage in vivo screening models, in an attempt to lower the number of candidate compounds proceeding to testing in mammalian models [1].”
Point 2: Materials and Methods – the authors should give the source of both clinical strains of bacteria (S. epidermidis HJ014 and MRSA).
Response 2: Thank you very much for seeing this. It has been a big omission that we forgot to write about the origin of these strains. The source of both strains has been added now in section 2.2:
2.2. Bacteria and growth conditions
For G. mellonella infection experiments four bacterial strains were used; Pseudomonas aeruginosa ATCC27853, Staphylococcus aureus ATCC29213, Escherichia coli ATCC25922, Staphylococcus epidermidis clinical strain (HJ056) [21] and a methicillin-resistant S. aureus (WKZ-2) [22]. All strains were cultivated in Mueller-Hinton broth (MHB) (BD Difco, DF0757-07-8) or agar (MHB with 1.5% agar). Cultures were incubated at 37°C with shaking at 180 rpm.
Point 3: Moreover, in this part of the manuscript authors omitted the research aiming in investigation of stability of antibiotic – they only presented it in the paragraph “Results”.
Response 3: Thank you very much it was a major mistake that we did not include this part in the methodology. Section “2.5. In vivo stability of antibiotic” is now added:
2.5. In vivo stability of antibiotic
To investigate the in vivo stability of antibiotic, healthy larvae of 350-400 mg were injected with 490 µg/ml ciprofloxacin (Sigma, 17850). Larvae injected with PBS served as control. Larvae were incubated at 37°C and at different time points (10 minutes, 1, 3 and 24 hours) haemolymph was extracted (individually for each larva), as described in section 2.4. Ten microliters of the freshly extracted haemolymph of ciprofloxacin- or PBS-injected larvae were placed on MH agar, for each time point, and were allowed to dry. One-hundred microliters of an overnight E. coli culture were spread over the agar, to create a lawn of bacteria. The plates were incubated overnight at 37°C and inhibition zones were observed the next day. The experiment was repeated twice, and representative plates are presented.
Point 4: I also would be grateful for a short information (one sentence) how the authors performed counting of bacteria.
Response 4: More information about bacteria counting has been added, in section “2.4. Bacterial load in the haemolymph”, lines 90-93:
“…Immediately after, serial 10-fold dilutions of the haemolymph were prepared in PBS. Ten microliters of each dilution were plated on MH agar, in triplicates, and were allowed to dry. Colonies were counted manually the next day, after overnight incubation at 37°C, and the CFU/ml haemolymph were calculated. No colonies were obtained from control, non-infected larvae.
Point 5: Conclusions – the authors should present short but separated conclusion.
Response 5: Thank you very much for this comment. It was a big omission that we did not write a conclusion section. Conclusion has been added (lines 189-197): “G. mellonella larvae are a reliable, cheap and immunologically complex model for the preliminary in vivo investigation of therapeutic potential of novel antimicrobial compounds, before proceeding to mammalian models. However, the outcomes of this model are highly influenced by the methodology and larva age, therefore an attempt should be made to standardize the research practices for this model. In this work, we presented a linear regression curve that correlates larva total weight with liquid volume. This curve can be used as a tool by G. mellonella model researchers in order to estimate the in vivo concentration of pathogens and administered compounds and will facilitate the comparison of results from different studies.”